# Neural Multi-Hop Reasoning With Logical Rules on Biomedical Knowledge Graphs

Yushan Liu[1,3], Marcel Hildebrandt[1,3], Mitchell Joblin[1], Martin Ringsquandl[1], Rime Raissouni[2,3], Volker Tresp[1,3]

[1] Siemens, Otto-Hahn-Ring 6, 81739 Munich, Germany
{firstname.lastname}@siemens.com
[2] Siemens Healthineers, Hartmannstraße 16, 91052 Erlangen, Germany
[3] Ludwig Maximilian University of Munich, Geschwister-Scholl-Platz 1, 80539 Munich, Germany

**Abstract.** Biomedical knowledge graphs permit an integrative computational approach to reasoning about biological systems. The nature of biological data leads to a graph structure that differs from those typically encountered in benchmarking datasets. To understand the implications this may have on the performance of reasoning algorithms, we conduct an empirical study based on the real-world task of drug repurposing. We formulate this task as a link prediction problem where both compounds and diseases correspond to entities in a knowledge graph. To overcome apparent weaknesses of existing algorithms, we propose a new method, PoLo, that combines policy-guided walks based on reinforcement learning with logical rules. These rules are integrated into the algorithm by using a novel reward function. We apply our method to Hetionet, which integrates biomedical information from 29 prominent bioinformatics databases. Our experiments show that our approach outperforms several state-of-the-art methods for link prediction while providing interpretability.

**Keywords:** Neural multi-hop reasoning · Reinforcement learning · Logical rules · Biomedical knowledge graphs

## 1 Introduction

Advancements in low-cost high-throughput sequencing, data acquisition technologies, and compute paradigms have given rise to a massive proliferation of data describing biological systems. This new landscape of available data spans a multitude of dimensions, which provide complementary views on the structure of biological systems. Historically, by considering single dimensions (i. e., single types of data), researchers have made progress in understanding many important phenomena. More recently, there has been a movement to develop statistical and computational methods that leverage more holistic views by simultaneously considering multiple types of data [40]. To achieve this goal, graph-based knowledge representation has emerged as a promising direction since the inherent flexibility of graphs makes them particularly well-suited for this problem setting.

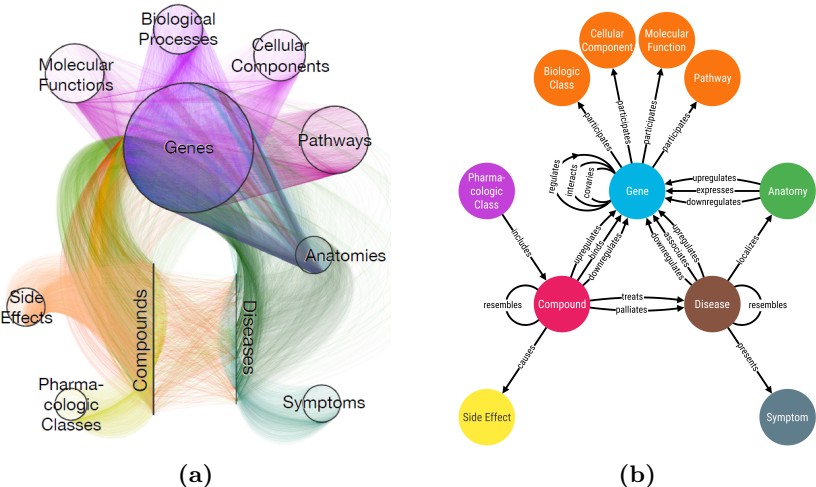

**Fig. 1.** (a) Visualization of the heterogeneous biomedical KG Hetionet [13][4]. (b) Schema of Hetionet: Hetionet has 11 different entity types and 24 possible relations between them. Source: `https://het.io/about/`.

Biomedical knowledge graphs (KGs) are becoming increasingly popular for tasks such as personalized medicine, predictive diagnosis, and drug discovery [9]. Drug discovery, for example, requires a multitude of biomedical data types combined with knowledge across diverse domains (including gene-protein bindings, chemical compounds, and biological pathways). These individual types of data are typically scattered across disparate data sources, published for domain-specific research problems without considering mappings to open standards. To this end, KGs and Semantic Web technologies are being applied to model ontologies that combine knowledge and integrate data contained in biomedical data sources, most notably Bio2RDF [2], for classical query-based question answering.

From a machine learning perspective, reasoning on biomedical KGs presents new challenges for existing approaches due to the unique structural characteristics of the KGs. One challenge arises from the highly coupled nature of entities in biological systems that leads to many high-degree entities that are themselves densely linked. For example, as illustrated in Figure 1a, genes interact abundantly among themselves. They are involved in a diverse set of biological pathways and molecular functions and have numerous associations with diseases.

A second challenge is that reasoning about the relationship between two entities often requires information beyond second-order neighborhoods [13]. Methods that rely on shallow node embeddings (e. g., TransE [4], DistMult [38]) typically do not perform well in this situation. Approaches that take the entire multi-hop neighborhoods into account (e. g., graph convolutional networks, R-GCN [30]) often have diminishing performance beyond two-hop neighborhoods (i. e., more

---

[4] Reprint under the use of the CC BY 4.0 license.

than two convolutional layers), and the high-degree entities can cause the aggregation operations to smooth out the signal [16]. Symbolic approaches (e. g., AMIE+ [10], RuleN [21]) learn logical rules and employ them during inference. These methods might be able to take long-range dependencies into account, but due to the massive scale and diverse topologies of many real-world KGs, combinatorial complexity often prevents the usage of symbolic approaches [14]. Also, logical inference has difficulties handling noise in the data [24].

Under these structural conditions, path-based methods present a seemingly ideal balance for combining information over multi-hop neighborhoods. The key challenge is to find meaningful paths, which can be computationally difficult if the search is not guided by domain principles. Our goal is to explore how a path-based approach performs in comparison with alternative state-of-the art methods and to identify a way of overcoming weaknesses present in current approaches.

We consider the drug repurposing problem, which is characterized by finding novel treatment targets for existing drugs. Available knowledge about drug-disease-interactions can be exploited to reduce costs and time for developing new drugs significantly. A recent example is the repositioning of the drug remdesivir for the novel disease COVID-19. We formulate this task as a link prediction problem where both compounds and diseases correspond to entities in a KG.

We propose a neuro-symbolic reasoning approach, PoLo (**Po**licy-guided walks with **Lo**gical rules), that leverages both representation learning and logic. Inspired by existing methods [5,12,18], our approach uses reinforcement learning to train an agent to conduct policy-guided random walks on a KG. As a modification to approaches based on policy-guided walks, we introduce a novel reward function that allows the agent to use background knowledge formalized as logical rules, which guide the agent during training. The extracted paths by the agent act as explanations for the predictions. Our results demonstrate that existing methods are inadequately designed to perform ideally in the unique structural characteristics of biomedical data. We can overcome some of the weaknesses of existing methods and show the potential of neuro-symbolic methods for the biomedical domain, where interpretability and transparency of the results are highly relevant to facilitate the accessibility for domain experts. In summary, we make the following contributions:

- We propose the neuro-symbolic KG reasoning method PoLo that combines policy-guided walks based on reinforcement learning with logical rules.
- We conduct an empirical study using a large biomedical KG where we compare our approach with several state-of-the-art algorithms.
- The results show that our proposed approach outperforms state-of-the-art alternatives on a highly relevant biomedical prediction task (drug repurposing) with respect to the metrics hits@$k$ for $k \in \{1, 3, 10\}$ and the mean reciprocal rank.

We briefly introduce the notation and review the related literature in Section 2. In Section 3, we describe our proposed method[5]. Section 4 details an experimental study, and we conclude in Section 5.

---

[5] The source code is available at `https://github.com/liu-yushan/PoLo`.

## 2    Background

### 2.1    Knowledge Graphs

Let $\mathcal{E}$ denote the set of entities in a KG and $\mathcal{R}$ the set of binary relations. Elements in $\mathcal{E}$ correspond to biomedical entities including, e. g., chemical compounds, diseases, and genes. We assume that every entity belongs to a unique type in $\mathcal{T}$, defined by the mapping $\tau : \mathcal{E} \rightarrow \mathcal{T}$. For example, $\tau(AURKC) = Gene$ indicates that the entity $AURKC$ has type $Gene$. Relations in $\mathcal{R}$ specify how entities are connected. We define a KG as a collection of triples $\mathcal{KG} \subset \mathcal{E} \times \mathcal{R} \times \mathcal{E}$ in the form $(h, r, t)$, which consists of a head entity, a relation, and a tail entity. Head and tail of a triple are also called source and target, respectively. From a graphical point of view, head and tail entities correspond to nodes in the graph while the

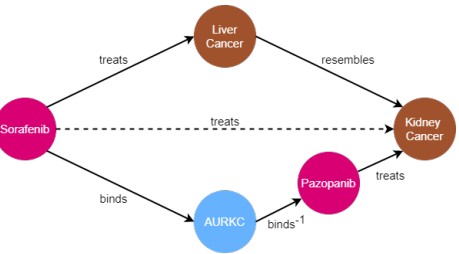

**Fig. 2.** Subgraph of Hetionet illustrating the drug repurposing use case. The two paths that connect the chemical compound sorafenib and the disease kidney cancer can be used to predict a direct edge between the two entities.

relation indicates the type of edge between them. For any relation $r \in \mathcal{R}$, we denote the corresponding inverse relation with $r^{-1}$ (i. e., $(h, r, t)$ is equivalent to $(t, r^{-1}, h)$). Triples in $\mathcal{KG}$ are interpreted as true known facts. For example, the triple $(Sorafenib, treats, Liver Cancer) \in \mathcal{KG}$ in Figure 2 corresponds to the known fact that the kinase inhibitor drug sorafenib is approved for the treatment of primary liver cancer. The $treats$ relation is of particular importance for this work since we frame the task of drug repurposing as a link prediction problem with respect to edges of the type $treats$. The domain of $treats$ consists of chemical compounds, and the range is given by the set of all diseases.

We further distinguish between two types of paths: instance paths and metapaths. An instance path of length $L \in \mathbb{N}$ on $\mathcal{KG}$ is given by a sequence

$$\left(e_1 \xrightarrow{r_1} e_2 \xrightarrow{r_2} \ldots \xrightarrow{r_L} e_{L+1}\right),$$

where $(e_i, r_i, e_{i+1}) \in \mathcal{KG}$. We call the corresponding sequence of entity types

$$\left(\tau(e_1) \xrightarrow{r_1} \tau(e_2) \xrightarrow{r_2} \ldots \xrightarrow{r_L} \tau(e_{L+1})\right)$$

a metapath. For example,

$$\left(Sorafenib \xrightarrow{treats} Liver\ Cancer \xrightarrow{resembles} Kidney\ Cancer\right)$$

constitutes an instance path of length 2, where

$$\left(Compound \xrightarrow{treats} Disease \xrightarrow{resembles} Disease\right)$$

is the corresponding metapath.

## 2.2   Logical Rules

Logical rules that are typically employed for KG reasoning can be written in the form $head \leftarrow body$. We consider cyclic rules of the form

$$(\tau_1, r_{L+1}, \tau_{L+1}) \leftarrow \bigwedge_{i=1}^{L} (\tau_i, r_i, \tau_{i+1}) \, ,$$

where $\tau_i \in \mathcal{T}$. The rule is called cyclic since the rule head (not to be confused with the head entity in a triple) connects the source $\tau_1$ and the target $\tau_{L+1}$ of the metapath $(\tau_1 \xrightarrow{r_1} \tau_2 \xrightarrow{r_2} \dots \xrightarrow{r_L} \tau_{L+1})$, which is described by the rule body. The goal is to find instance paths where the corresponding metapaths match the rule body in order to predict a new relation between the source and the target entity of the instance path. For the drug repurposing task, we only consider rules where the rule head is a triple with respect to the *treats* relation.
Define $CtD := (Compound, treats, Disease)$. Then, a generic rule has the form

$$CtD \leftarrow \left( Compound \xrightarrow{r_1} \tau_2 \xrightarrow{r_2} \tau_3 \xrightarrow{r_3} \dots \xrightarrow{r_L} Disease \right) \, .$$

In particular, the rule body corresponds to a metapath starting at a compound and terminating at a disease. For example (see Figure 2), consider the rule

$$CtD \leftarrow (Compound \xrightarrow{binds} Gene \xrightarrow{binds^{-1}} Compound \xrightarrow{treats} Disease) \, .$$

The metapath of the instance path

$$(Sorafenib \xrightarrow{binds} AURKC \xrightarrow{binds^{-1}} Pazopanib \xrightarrow{treats} Kidney\ Cancer)$$

matches the rule body, suggesting that sorafenib can also treat kidney cancer.

## 2.3   Related Work

Even though real-world KGs contain a massive number of triples, they are still expected to suffer from incompleteness. Therefore, link prediction (also known as KG completion) is a common reasoning task on KGs. Many classical artificial intelligence tasks such as recommendation problems or question answering can be rephrased in terms of link prediction.

Symbolic approaches have a far-reaching tradition in the context of knowledge acquisition and reasoning. Reasoning with logical rules has been addressed in areas such as Markov logic networks (MLNs) [28] or inductive logic programming [25]. However, such techniques typically do not scale well to modern, large-scale KGs. Recently, novel methods such as RuleN [21] and its successor AnyBURL [19,20] have been proposed that achieve state-of-the-art performance on popular benchmark datasets such as FB15k-237 [31] and WN18RR [7].

Subsymbolic approaches map nodes and edges in KGs to low-dimensional vector representations known as embeddings. Then, the likelihood of missing triples is approximated by a classifier that operates on the embedding space. Popular

embedding-based methods include translational methods like TransE [4], more generalized approaches such as DistMult[38] and ComplEx [32], multi-layer models like ConvE [7], and tensor factorization methods like RESCAL [26]. Moreover, R-GCN [30] and CompGCN [34] have been proposed, which extend graph convolutional networks [16] to multi-relational graphs. Despite achieving good results on the link prediction task, a fundamental problem is their non-transparent nature since it remains hidden to the user what contributed to the predictions. Moreover, most embedding-based methods have difficulties in capturing long-range dependencies since they only minimize the reconstruction error in the immediate first-order neighborhoods. Especially the expressiveness of long-tail entities might be low due to the small number of neighbors [11].

Neuro-symbolic methods combine the advantages of robust learning and scalability in subsymbolic approaches with the reasoning properties and interpretability of symbolic representation. For example, Neural LP [39] and Neural Theorem Provers (NTPs) [29] integrate logical rules in a differentiable way into a neural network architecture. The method pLogicNet [27] combines MLNs with embedding-based models and learns a joint distribution over triples, while the Logic Tensor Network [8] inserts background knowledge into neural networks in the form of logical constraints. However, many neuro-symbolic approaches suffer from limited transferability and computational inefficiency. Minervini et al. have presented two more scalable extensions of NTPs, namely the Greedy NTP (GNTP) [22], which considers the top-$k$ rules that are most likely to prove the goal instead of using a fixed set of rules, and the Conditional Theorem Prover (CTP) [23], which learns an adaptive strategy for selecting the rules.

Multi-hop reasoning or path-based approaches infer missing knowledge based on using extracted paths from the KG as features for various inference tasks. Along with a prediction, multi-hop reasoning methods provide the user with an explicit reasoning chain that may serve as a justification for the prediction. For example, the Path Ranking Algorithm (PRA) [17] frames the link prediction task as a maximum likelihood classification based on paths sampled from nearest neighbor random walks on the KG. Xiong et al. extend this idea and formulate the task of path extraction as a reinforcement learning problem (DeepPath [37]). Our proposed method is an extension of the path-based approach MINERVA [5], which trains a reinforcement learning agent to perform a policy-guided random walk until the answer entity to an input query is reached.

One of the drawbacks of existing policy-guided walk methods is that the agent might receive noisy reward signals based on spurious triples that lead to the correct answers during training but lower the generalization capabilities. Moreover, biomedical KGs often exhibit both long-range dependencies and high-degree nodes (see Section 4.1). These two properties and the fact that MINERVA's agent only receives a reward if the answer entity is correct make it difficult for the agent to navigate over biomedical KGs and extend a path in the most promising way. As a remedy, we propose the incorporation of known, effective logical rules via a novel reward function. This can help to denoise the reward signal and guide the agent on long paths with high-degree nodes.

## 3    Our Method

We pose the task of drug repurposing as a link prediction problem based on graph traversal. The general Markov decision process definition that we use has initially been proposed in the algorithm MINERVA [5], with our primary contribution coming from the incorporation of logical rules into the training process. The following notation and definitions are adapted to the use case. Starting at a query entity (a compound to be repurposed), an agent performs a walk on the graph by sequentially transitioning to a neighboring node. The decision of which transition to make is determined by a stochastic policy. Each subsequent transition is added to the current path and extends the reasoning chain. The stochastic walk process iterates until a finite number of transitions has been made. Formally, the learning task is modeled via the fixed-horizon Markov decision process outlined below.

*Environment* The state space $\mathcal{S}$ is given by $\mathcal{E}^3$. Intuitively, we want the state to encode the location $e_l$ of the agent for step $l \in \mathbb{N}$, the source entity $e_c$, and the target entity $e_d$, corresponding to the compound that we aim to repurpose and the target disease, respectively. Thus, a state $S_l \in \mathcal{S}$ for step $l \in \mathbb{N}$ is represented by $S_l := (e_l, e_c, e_d)$. The agent is given no information about the target disease so that the observed part of the state space is given by $(e_l, e_c) \in \mathcal{E}^2$. The set of available actions from a state $S_l$ is denoted by $\mathcal{A}_{S_l}$. It contains all outgoing edges from the node $e_l$ and the corresponding tail nodes. We also include self-loops for each node so that the agent has the possibility to stay at the current node. More formally, $\mathcal{A}_{S_l} := \{(r, e) \in \mathcal{R} \times \mathcal{E} : (e_l, r, e) \in \mathcal{KG}\} \cup \{(\emptyset, e_l)\}$. Further, we denote with $A_l \in \mathcal{A}_{S_l}$ the action that the agent performed in step $l$. The environment evolves deterministically by updating the state according to the previous action. The transition function is given by $\delta(S_l, A_l) := (e_{l+1}, e_c, e_d)$ with $S_l = (e_l, e_c, e_d)$ and $A_l = (r_l, e_{l+1})$.

*Policy* We denote the history of the agent up to step $l$ with $H_l := (H_{l-1}, A_{l-1})$ for $l \geq 1$, with $H_0 := e_c$ and $A_0 := \emptyset$. The agent encodes the transition history via an LSTM [15] by

$$\boldsymbol{h}_l = \text{LSTM}\left(\boldsymbol{h}_{l-1}, \boldsymbol{a}_{l-1}\right) , \tag{1}$$

where $\boldsymbol{a}_{l-1} := [\boldsymbol{r}_{l-1}; \boldsymbol{e}_l] \in \mathbb{R}^{2d}$ corresponds to the vector space embedding of the previous action (or the zero vector for $\boldsymbol{a}_0$), with $\boldsymbol{r}_{l-1}$ and $\boldsymbol{e}_l$ denoting the embeddings of the relation and the tail entity in $\mathbb{R}^d$, respectively. The history-dependent action distribution is given by

$$\boldsymbol{d}_l = \text{softmax}\left(\boldsymbol{A}_l\left(\boldsymbol{W}_2 \text{ReLU}\left(\boldsymbol{W}_1\left[\boldsymbol{h}_l; \boldsymbol{e}_l\right]\right)\right)\right) , \tag{2}$$

where the rows of $\boldsymbol{A}_l \in \mathbb{R}^{|\mathcal{A}_{S_l}| \times 2d}$ contain the latent representations of all admissible actions from $S_l$. The matrices $\boldsymbol{W_1}$ and $\boldsymbol{W_2}$ are learnable weight matrices. The action $A_l \in \mathcal{A}_{S_l}$ is drawn according to

$$A_l \sim \text{Categorical}\left(\boldsymbol{d}_l\right) . \tag{3}$$

The equations (1)–(3) are repeated for each transition step. In total, $L$ transitions are sampled, where $L$ is a hyperparameter that determines the maximum path length, resulting in a path denoted by

$$P := (e_c \xrightarrow{r_1} e_2 \xrightarrow{r_2} \ldots \xrightarrow{r_L} e_{L+1}) \, .$$

For each step $l \in \{1, 2, \ldots, L\}$, the agent can also choose to remain at the current location and not extend the reasoning path.

Equations (1) and (2) define a mapping from the space of histories to the space of distributions over all admissible actions. Thus, including Equation (3), a stochastic policy $\pi_\theta$ is induced, where $\theta$ denotes the set of all trainable parameters in equations (1) and (2).

*Metapaths* Consider the set of metapaths $\mathcal{M} = \{M_1, M_2, \ldots, M_m\}$, where each element corresponds to the body of a cyclic rule with *CtD* as rule head. For every metapath $M$, we denote with $s(M) \in \mathbb{R}_{>0}$ a score that indicates a quality measure of the corresponding rule, such as the confidence or the support with respect to making a correct prediction. Moreover, for a path $P$, we denote with $\tilde{P}$ the corresponding metapath.

*Rewards and optimization* During training, after the agent has reached its final location, a terminal reward is assigned according to

$$R(S_{L+1}) = \mathbb{1}_{\{e_{L+1}=e_d\}} + b\lambda \sum_{i=1}^{m} s(M_i)\mathbb{1}_{\tilde{P}=M_i} \, . \tag{4}$$

The first term indicates whether the agent has reached the correct target disease that can be treated by the compound $e_c$. It means that the agent receives a reward of 1 for a correct prediction. The second term indicates whether the extracted metapath corresponds to the body of a rule and adds to the reward accordingly. The hyperparameter $b$ can either be 1, i.e., the reward is always increased as long as the metapath corresponds to the body of a rule, or $b$ can be set to $\mathbb{1}_{\{e_{L+1}=e_d\}}$, i.e., an additional reward is only applied if the prediction is also correct. Heuristically speaking, we want to reward the agent for extracting a metapath that corresponds to a rule body with a high score. The hyperparameter $\lambda \geq 0$ balances the two components of the reward. For $\lambda = 0$, we recover the algorithm MINERVA.

We employ REINFORCE [35] to maximize the expected rewards. Thus, the agent's maximization problem is given by

$$\arg\max_\theta \mathbb{E}_{(e_c, treats, e_d) \sim \mathcal{D}} \, \mathbb{E}_{A_1, A_2, \ldots, A_L \sim \pi_\theta} \left[ R(S_{L+1}) \, \middle| \, e_c, e_d \right] \, , \tag{5}$$

where $\mathcal{D}$ denotes the true underlying distribution of $(e_c, treats, e_d)$-triples. During training, we replace the first expectation in Equation (5) with the empirical average over the training set. The second expectation is approximated by averaging over multiple rollouts for each training sample.

## 4    Experiments

### 4.1    Dataset Hetionet

Hetionet [13] is a biomedical KG that integrates information from 29 highly reputable and cited public databases, including the Unified Medical Language System (UMLS) [3], Gene Ontology [1], and DrugBank [36]. It consists of 47,031 entities with 11 different types and 2,250,197 edges with 24 different types. Figure 1b illustrates the schema and shows the different types of entities and possible relations between them.

Hetionet differs in many aspects from the standard benchmark datasets that are typically used in the KG reasoning literature. Table 1 summarizes the basic statistics of Hetionet along with the popular benchmark datasets FB15k-237 [31] and WN18RR [7]. One of the major differences between Hetionet and the two other benchmark datasets is the density of triples, i. e., the average node degree in Hetionet is significantly higher than in the other two KGs. Entities of type *Anatomy* are densely connected hub nodes, and in addition, entities of type *Gene* have an average degree of around 123. This plays a crucial role for our application since many relevant paths that connect *Compound* and *Disease* traverse entities of type *Gene* (see Figure 1b and Table 2). The total counts and the average node degrees according to each entity type are shown in Appendix A. We will discuss in Section 4.5 further how particularities of Hetionet impose challenges on existing KG reasoning methods.

We aim to predict edges with type *treats* between entities that correspond to compounds and diseases in order to perform candidate ranking according to the likelihood of successful drug repurposing in a novel treatment application. There are 1552 compounds and 137 diseases in Hetionet with 775 observed links of type *treats* between compounds and diseases. We randomly split these 755 triples into training, validation, and test set, where the training set contains 483 triples, the validation set 121 triples, and the test set 151 triples.

### 4.2    Metapaths as Background Information

Himmelstein et al. [13] evaluated 1206 metapaths that connect entities of type *Compound* with entities of type *Disease*, which correspond to various pharmacological efficacy mechanisms. They identified 27 effective metapaths that served as features for a logistic regression model that outputs a treatment probability of a compound for a disease. Out of these metapaths, we select the 10 metapaths as background information that have at most path length 3 and exhibit positive regression coefficients, which indicates their importance for predicting drug efficacy. We use the metapaths as the rule bodies and the confidence of the rules as the quality scores (see Section 3). The confidence of a rule is defined as the rule support divided by the body support in the data. We estimate the confidence score for each rule by sampling 5,000 paths whose metapaths correspond to the rule body and then computing how often the rule head holds. An overview of the 10 metapaths and their scores is given in Table 2.

**Table 1.** Comparison of Hetionet with the two benchmark datasets FB15k-237 and WN18RR.

| Dataset | Entities | Relations | Triples | Avg. degree |
|---|---|---|---|---|
| Hetionet | 47,031 | 24 | 2,250,197 | 95.8 |
| FB15k-237 | 14,541 | 237 | 310,116 | 19.7 |
| WN18RR | 40,943 | 11 | 93,003 | 2.2 |

**Table 2.** All 10 metapaths used in our model and their corresponding scores.

| $s(M)$ | Metapath $M$ |
|---|---|
| 0.446 | $(Compound \xrightarrow{includes^{-1}} Pharmacologic\ Class \xrightarrow{includes} Compound \xrightarrow{treats} Disease)$ |
| 0.265 | $(Compound \xrightarrow{resembles} Compound \xrightarrow{resembles} Compound \xrightarrow{treats} Disease)$ |
| 0.184 | $(Compound \xrightarrow{binds} Gene \xrightarrow{associates^{-1}} Disease)$ |
| 0.182 | $(Compound \xrightarrow{resembles} Compound \xrightarrow{treats} Disease)$ |
| 0.169 | $(Compound \xrightarrow{palliates} Disease \xrightarrow{palliates^{-1}} Compound \xrightarrow{treats} Disease)$ |
| 0.143 | $(Compound \xrightarrow{binds} Gene \xrightarrow{binds^{-1}} Compound \xrightarrow{treats} Disease)$ |
| 0.058 | $(Compound \xrightarrow{causes} Side\ Effect \xrightarrow{causes^{-1}} Compound \xrightarrow{treats} Disease)$ |
| 0.040 | $(Compound \xrightarrow{treats} Disease \xrightarrow{resembles} Disease)$ |
| 0.017 | $(Compound \xrightarrow{resembles} Compound \xrightarrow{binds} Gene \xrightarrow{associates^{-1}} Disease)$ |
| 0.004 | $(Compound \xrightarrow{binds} Gene \xrightarrow{expresses^{-1}} Anatomy \xrightarrow{localizes^{-1}} Disease)$ |

### 4.3   Experimental Setup

We apply our method PoLo to Hetionet and calculate the values for hits@1, hits@3, hits@10, and the mean reciprocal rank (MRR) for the link prediction task. All metrics in the paper are filtered [4] and evaluated for tail-sided predictions. During inference, a beam search is carried out to find the most promising paths, and the target entities are ranked by the probability of their corresponding paths. Moreover, we consider another evaluation scheme (PoLo (pruned)) that retrieves and ranks only those paths from the test rollouts that correspond to one of the metapaths in Table 2. All the other extracted paths are not considered in the ranking.

   We compare PoLo with the following baseline methods. The rule-based method AnyBURL [19,20] mines logical rules based on path sampling and uses them for inference. The methods TransE [4], DistMult [38], ComplEx [32], ConvE [6], and RESCAL [26] are popular embedding-based models, and we use the implementation from the LibKGE library[6]. To cover a more recent paradigm in graph-based machine learning, we include the graph convolutional approaches R-GCN [30] and CompGCN [34]. We also compare our method with

---

[6] https://github.com/uma-pi1/kge

the neuro-symbolic method pLogicNet [27]. The two neuro-symbolic approaches NTP [29] and Neural LP [39] yield good performance on smaller datasets but are not scalable to large datasets like Hetionet. We have also conducted experiments on the two more scalable extensions of NTP (GNTP [22] and CTP [23]), but both were not able to produce results in a reasonable time. More experimental details can be found in Appendix B.

### 4.4  Results

Table 3 displays the results for the experiments on Hetionet. The reported values for PoLo and MINERVA correspond to the mean across five independent training runs. The standard errors for the reported metrics are between 0.006 and 0.018. PoLo outperforms all baseline methods with respect to all evaluation metrics.

**Table 3.** Comparison with baseline methods on Hetionet.

| Method | Hits@1 | Hits@3 | Hits@10 | MRR |
|---|---|---|---|---|
| AnyBURL | 0.229 | 0.375 | 0.553 | 0.322 |
| TransE | 0.099 | 0.199 | 0.444 | 0.205 |
| DistMult | 0.185 | 0.305 | 0.510 | 0.287 |
| ComplEx | 0.152 | 0.285 | 0.470 | 0.250 |
| ConvE | 0.100 | 0.225 | 0.318 | 0.180 |
| RESCAL | 0.106 | 0.166 | 0.377 | 0.187 |
| R-GCN | 0.026 | 0.245 | 0.272 | 0.135 |
| CompGCN | 0.172 | 0.318 | 0.543 | 0.292 |
| pLogicNet | 0.225 | 0.364 | 0.523 | 0.333 |
| MINERVA | 0.264 | 0.409 | 0.593 | 0.370 |
| PoLo | 0.314 | 0.428 | 0.609 | 0.402 |
| PoLo (pruned) | **0.337** | **0.470** | **0.641** | **0.430** |

Applying the modified ranking scheme, our method yields performance gains of 27.7% for hits@1, 14.9% for hits@3, 8.1% for hits@10, and 16.2% for the MRR with respect to best performing baseline.

Figure 3a shows the rule accuracy, i. e., the percentage of correct target entities for extracted paths that follow rule metapaths, for PoLo and MINERVA during training. Both lines behave similarly in the beginning, but the rule accuracy of PoLo increases significantly around epoch 20 compared to MINERVA. It seems that giving the agent an extra reward for extracting rules also improves the probability of arriving at correct target entities when applying the rules. We also compare the metric hits@1 (pruned) for the evaluation of the validation set during training (see Figure 3b). Around epoch 20, where the rule accuracy of PoLo increases compared to MINERVA, hits@1 (pruned) also increases while it decreases for MINERVA. The additional reward for extracting rule paths could

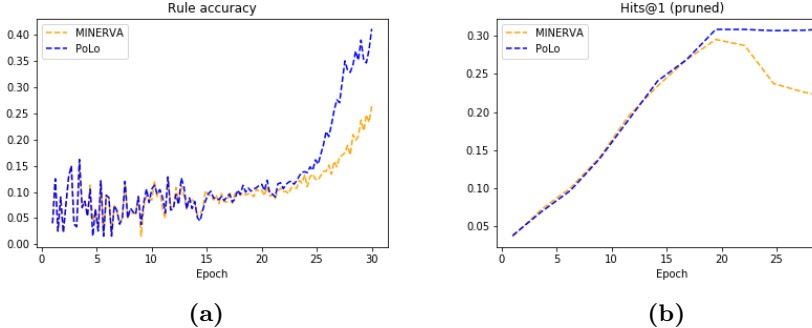

**Fig. 3.** (a) Rule accuracy during training. (b) Hits@1 (pruned) for the evaluation of the validation set during training.

be seen as a regularization that alleviates overfitting and allows for longer training for improved results.

The metapath that was most frequently extracted by PoLo during testing is

$$(\textit{Compound} \xrightarrow{\textit{causes}} \textit{Side Effect} \xrightarrow{\textit{causes}^{-1}} \textit{Compound} \xrightarrow{\textit{treats}} \textit{Disease})\,.$$

This rule was followed in 37.3% of the paths during testing, of which 16.9% ended at the correct entity.

During testing, PoLo extracted metapaths that correspond to rules in 41.7% of all rollouts while MINERVA only extracted rule paths in 36.9% of the cases. The accuracy of the rules, i. e., the percentage of correct target entities when rule paths are followed, is 19.0% for PoLo and 17.6% for MINERVA.

### 4.5   Discussion

We have integrated logical rules as background information via a new reward mechanism into the multi-hop reasoning method MINERVA. The stochastic policy incorporates the set of rules that are presented to the agent during training. Our approach is not limited to MINERVA but can act as a generic mechanism to inject domain knowledge into reinforcement learning-based reasoning methods on KGs [18,37]. While we employ rules that are extracted in a data-driven fashion, our method is agnostic towards the source of background information.

The additional reward for extracting a rule path can be considered as a regularization that induces the agent to walk along metapaths that generalize to unseen instances. In particular, for PoLo (pruned), we consider only extracted paths that correspond to the logical rules. However, the resulting ranking of the answer candidates is not based on global quality measures of the rules (e. g., the confidence). Rather, the ranking is given by the policy of the agent (i. e., metapaths that are more likely to be extracted are ranked higher), which creates an adaptive reweighting of the extracted rules that takes the individual instance paths into account.

Multi-hop reasoning methods contain a natural transparency mechanism by providing explicit inference paths. These paths allow domain experts to evaluate and monitor the predictions. Typically, there is an inherent trade-off between explainability and performance, but surprisingly, our experimental findings show that path-based reasoning methods outperform existing black-box methods on the drug repurposing task. Concretely, we compared our approach with the embedding-based methods TransE, DistMult, ComplEx, ConvE, and RESCAL. These methods are trained to minimize the reconstruction error in the immediate first-order neighborhood while discarding higher-order proximities. However, most explanatory metapaths in the drug repurposing setting have length 2 or more [13]. While MINERVA and PoLo can explicitly reason over multiple hops, our results indicate that embedding-based methods that fit low-order proximities seem not to be suitable for the drug repurposing task, and it is plausible that other reasoning tasks on biomedical KGs could result in similar outcomes.

R-CGN and CompGCN learn node embeddings by aggregating incoming messages from neighboring nodes and combining this information with the node's own embedding. These methods are in principle capable of modeling long-term dependencies. Since the receptive field contains the entire set of nodes in the multi-hop neighborhood, the aggregation and combination step essentially acts as a low-pass filter on the incoming signals. This can be problematic in the presence of many high-degree nodes like in Hetionet where the center node receives an uninformative signal that smooths over the neighborhood embeddings.

The approaches pLogicNet and AnyBURL both involve the learning of rules and yield similar performance on Hetionet, which is worse than PoLo. Most likely, the large amount of high-degree nodes in Hetionet makes the learning and application of logical rules more difficult. Other neuro-symbolic methods such as NTP, its extensions, and Neural LP were not scalable to Hetionet.

To illustrate the applicability of our method, consider the example of the chemical compound sorafenib (see Figure 2), which is known for treating liver cancer, kidney cancer, and thyroid cancer. The top predictions of our model for new target diseases include pancreatic cancer, breast cancer, and hematologic cancer. This result seems to be sensible since sorafenib already treats three other cancer types. The database ClinicalTrials.gov [33] lists 16 clinical studies for testing the effect of sorafenib on pancreatic cancer, 33 studies on breast cancer, and 6 studies on hematologic cancer, showing that the predicted diseases are meaningful targets for further investigation. Another example of drug repurposing on Hetionet is provided in Appendix C.

### 4.6   Experiments on Other Datasets

We also conduct experiments on the benchmark datasets FB15k-237 and WN18RR and compare PoLo with the other baseline methods. Since we do not already have logical rules available, we use the rules learned by AnyBURL. We can only apply cyclic rules for PoLo, so we also compare to the setting where we only learn and apply cyclic rules with AnyBURL.

Our method mostly outperforms MINERVA and Neural LP on both datasets. For FB15k-237, PoLo has worse performance than AnyBURL and most embedding-based methods, probably because the number of unique metapaths that occur a large number of times in the graph is lower compared to other datasets [5]. This makes it difficult for PoLo to extract metapaths sufficiently often for good generalization. pLogicNet yields better performance on FB15k-237 than PoLo but worse performance on WN18RR. The results of AnyBURL on FB15k-237 and WN18RR when only using cyclic rules are worse than when also including acyclic rules. It seems that acyclic rules are important for predictions as well, but PoLo cannot make use of these rules. The detailed results for both datasets can be found in Appendix D.

## 5 Conclusion

Biomedical knowledge graphs present challenges for learning algorithms that are not reflected in the common benchmark datasets. Our experimental findings suggest that existing knowledge graph reasoning methods face difficulties on Hetionet, a biomedical knowledge graph that exhibits both long-range dependencies and a multitude of high-degree nodes. We have proposed the neuro-symbolic approach PoLo that leverages both representation learning and logic. Concretely, we integrate logical rules into a multi-hop reasoning method based on reinforcement learning via a novel reward mechanism. We apply our method to the highly relevant task of drug repurposing and compare our approach with embedding-based, logic-based, and neuro-symbolic methods. The results indicate a better performance of PoLo compared to popular state-of-the-art methods. Further, PoLo also provides interpretability by extracting reasoning paths that serve as explanations for the predictions.

**Acknowledgements.** This work has been supported by the German Federal Ministry for Economic Affairs and Energy (BMWi) as part of the project RAKI (01MD19012C).

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
