# OpenReview forum: "Neural Multi-Hop Reasoning With Logical Rules on Biomedical Knowledge Graphs"
_eswc-conferences.org/ESWC/2021/Conference/Research_Track — ESWC 2021 Research_

### Official Review · AnonReviewer4 · 2021-01-01
**The authors studied the drug repurposing problem and proposed a neuro-symbolic KG reasoning approach.**

**Rating:** 1
**Confidence:** 3
**Impact:** 3
**Design And Technical Quality:** 3

**Review:**

In this paper, the authors studied the drug repurposing problem and proposed a neuro-symbolic KG reasoning approach. The key idea that underpins is to integrate metapaths (logical rules) into reinforcement learning. The proposed MINERVA+ outperformed several competitors on the biomedical datasets.

Generally, I have three main concerns that are listed as follows:

1. I noticed that the authors have published a very similar paper at ICML 2020 Workshop (Graph Representation Learning and Beyond). I roughly compared the two works and cannot see the significant differences. So, I would ask the authors to clarify the new contributions for this submission.

2. Combining logical rules or metapaths and representation learning is often seen in other works. For example, in the recommender systems [1,2], some works used pre-defined metapaths while others mined the metapaths automatically. Compared with these works, the novelty of this work is relatively marginal.

[1] metapath2vec: Scalable representation learning for heterogeneous networks. KDD 2017

[2] Metapath-guided heterogeneous graph neural network for intent recommendation. KDD 2019

3. Although the experiment results showed the superiority of the proposed approach, I would like to see some concrete examples. Since the authors focused on the drug repurposing problem, have you found any specific cases? At least, can you find some cases that we have already know, such as Viagra? I can understand this is a very difficult problem, but the authors may show some positive evidence.

I acknowledge that I have read the above response. Particularly, the author clarified the differences between the previous version and the current submission. So I raise my score to weak accept.

**Anonymity:**

Yes, I would like my review to remain anonymous.

**Reuse And Availability:**

3: Medium

**Strong Points:**

S1. Clear writing.

S2. Potential impact on drug repurposing.

**Subreviewer:**

I submitted this review.

**Weak Points:**

W1. Overlap with the paper published in ICML workshop.

W2. The novelty of combining metapaths and deep learning is not new.

W3. Lack of concrete examples in real applications.

---

> ### Author Rebuttal · Authors · 2021-01-30
>
> Thank you very much for you feedback and valuable comments!
>
> **A1**: Our submission is a more comprehensive and revised version of the short paper previously published at the workshop. More specifically, the submitted paper contains the following extensions:
> - Related work section about symbolic, subsymbolic, and neuro-symbolic methods.
> - An improved reward function (see Equation (4)), adding the option to increase the reward if the extracted metapath corresponds to a rule body even if the target entity is not correct. This option is introduced to alleviate the problem of sparse rewards during training.
> - Extensive experimental evaluation with more baseline methods (new version of AnyBURL, DistMult, ComplEx, ConvE, CompGCN, and pLogicNet). We also conducted experiments on the neuro-symbolic method Neural LP, which was unfortunately not scalable to Hetionet.
> - More in-depth analysis of MINERVA+ and comparison to plain MINERVA.
> - Experimental results on the benchmark datasets FB15K-237 and WN18RR to examine the general capabilities and limitations of the approach.
>
> **A2**: The idea of combining metapaths and representation learning has indeed been explored before, while there are fundamental differences between the methods and applications introduced in the cited papers [1, 2] and our paper. The approach metapath2vec [1] formalizes metapath-based random walks and applies a skip-gram model for learning node embeddings for node classification, clustering, and similarity search. MEIRec [2] uses metapath-guided neighbors for the aggregation within a graph neural network for intent recommendation. Both metapath2vec and MEIRec are based on nearest neighbor-random walks in the KG that *must* follow the specified metapaths, which would be detrimental in the case of noisy metapaths. Our method induces the RL agent to follow predefined metapaths, whereas the agent also has the choice to walk along other paths. Furthermore, metapath2vec and MEIRec do not provide explanations for their predictions and are thus less interpretable for humans, while MINERVA+ outputs specific paths that lead to the predicted entities as explanations.
>
> Our method introduces a simple and general extension by employing logical rules via reward engineering. To the best of our knowledge, it is the first method that directly integrates symbolic logical rules into a neural path-based method based on reinforcement learning.
> The application in the biomedical domain is highly relevant due to the need for interpretability and transparency to facilitate the accessibility for domain experts. Our method meets the challenges of solving prediction tasks on biomedical KGs that include high-degree entities and long-term dependencies beyond second-order neighborhoods. Both these kinds of (biomedical) graphs and the task of drug repurposing are not commonly encountered in the context of link prediction, and our results show the potential of neuro-symbolic methods for the biomedical domain.
>
> [1] Dong, Y. et al. metapath2vec: scalable representation learning for heterogeneous networks. KDD 2017.
>
> [2] Fan, S. et al. Metapath-guided heterogeneous graph neural network for intent recommendation. KDD 2019.
>
> **A3**: We demonstrate an example at the end of Section 4.5, which involves the chemical compound sorafenib. Sorafenib (marketed as Nexavar) is a kinase inhibitor for treating liver cancer, kidney cancer, and thyroid cancer. The three highest predictions of our model for new target diseases include hematologic cancer, breast cancer, and Barrett’s esophagus. This result seems to be sensible since sorafenib already treats three other cancer types. The database ClinicalTrails.gov, which registers privately and publicly funded clinical studies conducted around the world, lists 4 completed clinical studies for testing the effect of sorafenib on hematologic cancer and one study for Barrett’s esophagus. For breast cancer, there are 14 completed studies and 4 active ones. It seems that the predicted diseases are meaningful targets for further investigation.
>
> Since repurposed drugs need to undergo rigorous clinical trials and known drug-disease interactions are already included in Hetionet, domain experts are needed to judge the medical appropriateness of new findings. The amount of clinical trials can act as an indication for the meaningfulness of the results. We will try to find more examples with positive evidence and add them to the supplementary material.

---

### Official Review · AnonReviewer3 · 2021-01-12
**An interesting an novel approach that deserves to be published**

**Rating:** 3
**Confidence:** 4
**Impact:** 4
**Design And Technical Quality:** 5

**Review:**

The paper addresses the problem of link prediction in Biological Knowledge Graphs (KG) where two entities can be semantically related through a complex path. For example, it is possible to discover that a compound is able to treat a disease by navigating a proper path in the KG. However, current embedding methods are not able to perform such a complex inference. The authors therefore propose a novel approach, called MINERVA+, based on multi-hop reasoning in a reinforcement learning where the reward takes into account also background knowledge in form of rules extracted from the KG. Results on the standard Hetionet KG on link prediction overcome the state-of-the-art whereas the method has to be improved for other standard KGs. Nevertheless, the method is interesting and it is an important contribution to the AI community that integrates ML methods with symbolic knowledge.

**Anonymity:**

Yes, I would like my review to remain anonymous.

**Reuse And Availability:**

4: High

**Strong Points:**

- The paper is clear, well structured and well written;
- It addresses an important problem;
- I appreciate the way of posing the main problems followed by the proposed solutions;
- The method is simple and quite general. Therefore, it can be applied to other domains where reinforcement learning can be enhanced with background knowledge.
- Even if MINERVA+ does not improve the state-of-the-art on FB and WN datasets it is an interesting Neural-Symbolic method. It is able to discover new diseases that can be treated with well known compounds.

**Subreviewer:**

I submitted this review.

**Weak Points:**

I would state/discuss more neural symbolic methods in Section 2.2. For example, also Logic Tensor Networks [1] adds background knowlege as regularization term. In addition, Logic Tensor Networks has also been applied to KGs problems [1].

A couple of things are not clear to me and maybe a clarification in the paper would improve its comprehension.
- Section 4.3, what do you mean that all metrics are filtered[4]?
- Section 4.3, what do you use beam search for?
- A couple of examples about cyclic and acyclic rules would clarify section 4.6. The definition of head and body ending with the same entity is ambigous to me. In addition, I would state that MINERVA+ can use only cyclic rules in section 3 as this is a core property of the method.

Adding more examples at the end of Section 4.5 would make the proposal stronger even if it does not improve the other methods on FB and WN datasets.

A tiny typo: page 13, 8th line from below, Hetionet. where -> Hetionet, where

[1] Donadello, I., & Serafini, L. (2019, July). Compensating supervision incompleteness with prior knowledge in semantic image interpretation. In 2019 International Joint Conference on Neural Networks (IJCNN) (pp. 1-8). IEEE.

---

> ### Author Rebuttal · Authors · 2021-01-30
>
> Thank you very much for your suggestions and valuable comments!
>
> We agree that the paragraph on neural-symbolic methods in Section 2.2 is rather short. We will add a more elaborate description on neural-symbolic methods and also include Logic Tensor Network as a further neuro-symbolic approach that makes use of background knowledge.
>
> More examples would definitely strengthen the paper. Due to space limitations, we will include more examples in the supplementary material.
>
> **Q1**: Section 4.3, what do you mean that all metrics are filtered?
>
> **A1**: We follow the evaluation protocol introduced by Bordes et al. [1] for calculating the metrics hits@k and MRR. During inference, the candidate tail entities for the test query $(h, r, ?)$ are obtained from the RL agent’s extracted paths and ranked by decreasing probability of the paths. It is possible that true entities, which exist in the KG but are not the answer to the test query, are ranked higher than the unique answer to the test query. Since higher-ranked true entities should not negatively affect the ranking, the true entities that are not the correct answer are filtered out/removed from the list of candidates. The metrics obtained under this setting are commonly referred to as filtered.
>
> [1] Bordes, A. et al. Translating embeddings for modeling multi-relational data. NeurIPS 2013.
>
> **Q2**: Section 4.3, what do you use beam search for?
>
> **A2**: Beam search keeps track of the most promising paths during inference. Starting from the query subject, the $k$ most probable next neighbors are identified according to the stochastic policy learned during training. In the next step, the most probable neighbors are generated for each one-hop neighbor identified in the first step, and the resulting set of paths is reduced to the $k$ most promising ones. The steps are repeated until the paths have the desired length, and the target entities of the paths specify the answer candidates. The original MINERVA algorithm uses beam search during inference, and we keep the same setting to find the paths that lead to correct entities with high probability.
>
> **Q3**: A couple of examples about cyclic and acyclic rules would clarify Section 4.6. The definition of head and body ending with the same entity is ambiguous to me. In addition, I would state that MINERVA+ can use only cyclic rules in section 3 as this is a core property of the method.
>
> **A3**: Thanks for the suggestion. We will introduce cyclic and acyclic rules in Section 2.1 and add examples to make the concepts clearer to the reader. It is also a good idea to state already in Section 3 that MINERVA+ uses only cyclic rules.
>
> By *cyclic* rules we mean rules of the form
> $$
> h(e_1, e_{n+1}) \leftarrow \bigwedge_{i=1}^n r_i(e_{i}, e_{i+1}),
> $$
> where the body of the rule can be written as a metapath $(e_1 \xrightarrow{r_1} e_2 \xrightarrow{r_2} … \xrightarrow{r_n} e_{n+1})$. An atom $(e_i, r_i, e_{i+1})$ can be expressed as an $(h, r, t)$-triple $(e_i, r_i, e_{i+1})$ in the KG. The rule head connects the source $e_1$ and the target $e_{n+1}$ of the metapath described by the rule body. Since the body and the head form a cycle in the (undirected) KG, we call these rules cyclic rules. Rules of any other form for which head and body do not constitute a cycle in the KG are called acyclic.

---

### Official Review · AnonReviewer2 · 2021-01-14
**An interesting paper with some promising results**

**Rating:** 1
**Confidence:** 4
**Impact:** 3
**Design And Technical Quality:** 4

**Review:**

The paper explores the use of a novel path-based reasoning method for KG to perform multi-hop reasoning on biomedical KG, and evaluates its efficiency with respect to other existing symbolic, sub-symbolic and neuro-symbolic reasoning approaches for link prediction in KGs. The challenge presented  by biomedical KG is the long-range dependencies between entities and the high number of high-degree nodes. To address this challenge the paper proposes a neuro-symbolic approach that uses both representation learning and logical rules to solve link prediction tasks in biomedical KGs. The method uses reinforcement learning to lear best path walks over a given large KG, guided by a given set of logical rules that are domain-specific and relevant to the target prediction.The satisfiability of the body conditions in such rules during the path walk is used as a regularisation term in the reward function for the RL so that prefer his given to those branch selections that would maximise the satisfaction of the given meta-path (or rules). The results are promising as they seem to outperform existing methods for link prediction.

The paper addresses an important problem. It clearly outline the two main challenges of biomedical KG from a machine learning standpoint: the highly coupled nature of entities and the frequent requirement of information beyond second-order neighbours in solving akin prediction task, such as drug repurposing. It cites relevant literature and existing approaches for link predictions in general KGs covering the three areas of pure symbolic, pure sub-symbolic and neuro-symbolic. The evaluation is comprehensive too. as it relates the results on the drug  repurposing problem with respect to most of the key state-of-the-art systems.

I believe the paper provides a nice idea support by good evaluation results.


**Anonymity:**

Yes, I would like my review to remain anonymous.

**Reuse And Availability:**

3: Medium

**Strong Points:**

The paper is well written and well structured. The choice of neuro-symbolic method for solving the problem is appropriate as it enables to maintain the interpretability of the results, which is very important in the biomedical domain for at the uptake of ML technologies by domain experts. There are aware some concerns, highlighted below, that the authors should comment upon.

**Subreviewer:**

I submitted this review.

**Weak Points:**

I have a three main concerns that I believe the authors should address.

Firstly, the use of logical rules, as background domain-specific knowledge to  guide the RL search, is interesting. However, the authors should comment on the provenance of these rules in real practice. If these are rules defined by experts I would expect them have a reasonably short conjunction of relations in the body.  So it is not clear how they would help in situation where link predictions depend upon long chains. It would be good if the authors could comment on the length of the rules that are expected in biomedical domains and the length of instance paths in current biomedical KGs.

Secondly, domain-specific knowledge expressed as rules by domain experts often come with a reacher language in particular which uses negation as part of the body conditions. How generalisable is their approach to more expressive rule-based languages? Also what would it mean for a meta-path rule definitions to have negated conditions?

Thirdly, meta-path rules could be expected to be automatically extracted from KGs in order to be then used at inference time to some a prediction task. This seems to have been the case for the evaluation over the FB15k- 237 and WN18RR KGs since logical rules from domain experts in these domains were not available. So the questions is how sensitive the  method is to the level of accuracy of these learned rules? It would have been nice to see an evaluation of the performance of the proposed approach conditional to percentage of noise over the rules used by the RL.

---

> ### Author Rebuttal · Authors · 2021-01-30
>
> Thank you very much for your feedback and valuable comments!
>
> **A1**: In real practice, the rules could either be handcrafted by domain experts or mined from given data. In Hetionet, the shortest path length between compounds and diseases varies between 2 and 3, while there exist substantially more longer paths between compounds and diseases that are relevant for the link prediction problem. We consider rule bodies of length 3 already as long since many embedding-based methods have difficulties in capturing long-range dependencies beyond second-order neighborhoods. Our experiments show that already rules up to length 3 are sufficient to improve the results on MINERVA+ compared to plain MINERVA. We expect domain experts to be able to provide rules up to length 3, especially rules that contain recurrent formulations, e.g.,
>
> [$C$ stands for $\textit{Compound}$ and $D$ for $\textit{Disease}$]
> $$
> \textit{CtD} \leftarrow (\textit{C} \xrightarrow{\textit{resembles}} \textit{C} \xrightarrow{\textit{resembles}} ...  \xrightarrow{\textit{resembles}}\textit{C} \xrightarrow{\textit{treats}} \textit{D}).
> $$
> A further possibility to support the experts in creating rules would be to generate longer metapaths based on the KG schema, approximate the confidence of each rule, and show the most promising candidates to the experts who can then choose which rules to integrate for subsequent training.
>
> **A2**: Our method integrates rules of the form
> $$
> h(e_1, e_{n+1}) \leftarrow \bigwedge\limits_{i=1}^n r_i(e_{i}, e_{i+1}),
> $$ where the body of the rule can be written as a metapath
> $
> (e_1  \xrightarrow{r_1} e_2  \xrightarrow{r_2} …  \xrightarrow{r_n} e_{n+1}).
> $
> An atom $r_i(e_i, e_{i+1})$ can be expressed as an $(h, r, t)$-triple $(e_i, r_i, e_{i+1})$ in the KG. The rule head connects the source $e_1$ and the target $e_{n+1}$ of the metapath described by the rule body. Since the body and the head form a cycle in the (undirected) KG, we call these rules *cyclic* rules.
>
> Given an exemplary cyclic rule
> $$
> CtD \leftarrow (C_1  \xrightarrow{resembles} C_2  \xrightarrow{treats} D),
> $$
> we could add a (negated) body atom $(C_2  \xrightarrow{\textit{does not cause}} \textit{(specific) Side Effect})$ indicating that the second compound in the path must not cause any (specific) side effect.
> We can check the new condition by using the information from the KG and assign the reward for the RL agent depending on the truth value of the whole rule body. Note that we have to make a closed-world assumption on the KG to justify these negations (negation as failure).
>
> If we negate a predicate that is part of the rule metapath, e.g.,
> $$
> CtD \leftarrow (C \xrightarrow{resembles} C  \xrightarrow{\textit{does not treat}} D),
> $$
> it would mean that the second edge of the path must not connect a compound and a disease. The target of the path might not be a disease so that a treatment prediction using this rule would be meaningless in our setting. Unless we modify the path sampling mechanism, negations cannot be directly incorporated. However, this is an interesting direction for future work.
>
> In summary, we can extend the rules by including expressions from a richer rule-based language as long as it is feasible to check the new expressions in the KG. Further, the rule body has to contain a metapath that, together with the rule head, constitutes a cycle in the KG.
>
> **A3**: As a probabilistic approach that relies on entity/relation embeddings and a stochastic transition policy, MINERVA+ is able to cope with noise and uncertainty in the data and logical rules better than purely rule-based methods. The confidence of a rule reflects the accuracy, and a rule with low confidence has less influence on the rule-dependent additional reward, which is a linear function of the confidence (see Equation (4)). In the case when the additional reward is only applied if the target entity is correct ( $b = I_{\{e_{T+1} = e_d\}}$), rules with perhaps wrong confidence values even do not affect the training at all. Rare metapaths that are seldom extracted by the agent also affect the training to a lesser extent even if they have a high confidence. Generally, plain MINERVA is able to find reasonable (rule) metapaths by rewarding the agent if the target entity of the path is correct. Since we do not force the agent to walk along rule metapaths and also reward the agent for ending at the correct entity, the method can adapt the importance (through embeddings/stochastic policy) it assigns to  rules with confidence values that deviate from the true confidences. We agree that it would be interesting to have an even more in-depth analysis of reasoning over noisy KGs to explore strengths and weakness of each method, but we believe that this would best be positioned in a separate paper.

---

### Official Review · AnonReviewer1 · 2021-01-14
**Considering a very important topic with some possibilities for improvement**

**Rating:** 1
**Confidence:** 4
**Impact:** 4
**Design And Technical Quality:** 4

**Review:**

## Significance

- the approach of incorporating logical rules in the training process is certainly addressing one of the big current topics of our field

## Quality

- the evaluation is clear and extensive (as is the rest of the paper, discussed below)

- could you give some references for particular claims (which are reasonable, but should be corroborated)

  - "these methods might be able to take long-range dependencies into account, but due to the massive scale and diverse topologies of many real-world KGs, combinatorial com-plexity often prevents the usage of symbolic approaches."

  - "Also, logical inference has difficulties handling noise in the data."

  - "Moreover, most embedding-based methods can-not capture the compositionality expressed by long reasoning chains. This often limits their applicability to complex reasoning tasks."

-  maybe I overlooked something, but I could not find the link to the source code in either the paper or the supplementary material. Given that this is promised, I do not let this affect the rating.

## Clarity

- the paper is well-structured, including a clear introduction, but also throughout

## Originality

- an insightful comparison of benchmark datasets is provided

- not many arguments are given corroborating the depth of contribution of the paper, and thus its significance and originality



**Anonymity:**

Yes, I would like my review to remain anonymous.

**Reuse And Availability:**

4: High

**Strong Points:**

- for significance, the approach of incorporating logical rules in the training process is certainly addressing one of the big current topics of our field

- for clarity, the paper is well-structured, including a clear introduction, but also throughout

- for originality, an insightful comparison of benchmark datasets is provided

- for quality, the evaluation is clear and extensive


**Subreviewer:**

I submitted this review.

**Weak Points:**


- for significance and originality, not many arguments are given corroborating the depth of contribution of the paper, and thus its significance and originality
  (perhaps this is the biggest weak point)

- for quality and clarity, could you give some references for particular claims (which are reasonable, but should be corroborated)

  - "these methods might be able to take long-range dependencies into account, but due to the massive scale and diverse topologies of many real-world KGs, combinatorial com-plexity often prevents the usage of symbolic approaches."

  - "Also, logical inference has difficulties handling noise in the data."

  - "Moreover, most embedding-based methods can-not capture the compositionality expressed by long reasoning chains. This often limits their applicability to complex reasoning tasks."

- for quality, maybe I overlooked something, but I could not find the link to the source code in either the paper or the supplementary material. Given that this is promised, I do not let this affect the rating.

---

> ### Author Rebuttal · Authors · 2021-01-30
>
> Thank you very much for your feedback and valuable comments!
>
> **Q1**: For significance and originality, not many arguments are given corroborating the depth of contribution of the paper, and thus its significance and originality.
>
> **A1**: Our contributions concern two directions: the first one refers to the algorithmic side, and the second one involves the application to a new problem domain and its extensive evaluation. The idea is to introduce a simple but elegant extension by employing logical rules via reward engineering to make use of domain knowledge while providing better support for path-based explanations. To the best of our knowledge, our method is the first approach that directly integrates symbolic logical rules into a neural multi-hop reasoning method based on reinforcement learning. The application in the biomedical domain is highly relevant due to the need for interpretability and transparency to facilitate the accessibility for domain experts. Our method meets the challenges of solving prediction tasks on biomedical KGs that include high-degree entities and long-term dependencies beyond second-order neighborhoods. Both these kinds of (biomedical) graphs and the task of drug repurposing are not commonly encountered in the context of link prediction, and our results show the potential of neuro-symbolic methods for the biomedical domain. . To make the impact and originality of our contributions more apparent, we will emphasize the above arguments more in the introduction section of the paper.
>
> **Q2**: For quality and clarity, could you give some references for particular claims (which are reasonable, but should be corroborated)?
>
> **A2**: Thanks for the comment. We will be more precise in the paper and include references to underpin the claims.
>
> **(i)** Generally, sound and complete reasoning over linked data suffers from high computational complexity [1]. Both symbolic and neuro-symbolic algorithms tend to be affected by computational limitations when applied to large-scale datasets. For example, mining rules of length 3 with AMIE+ [2] on the dataset YAGO2 takes more than 2 days when taking type constraints into account [3]. The neuro-symbolic method Neural LP [4] does not scale to the dataset NELL-995 [5], while Neural Theorem Prover [6], another neuro-symbolic approach, does not scale to any of the datasets WN18RR, FB15k-237, and NELL-995 [4, 5]. It is possible to improve computational efficiency by, e.g., resorting to sampling techniques [7] or ignoring type constraints [2].
>
> **(ii)** Logic programming is often applied to learn human-readable if-then rules, e.g., first-order Horn clauses. Methods in the field of (inductive) logic programming often perform worse on noisy and ambiguous data since errors in the training data could produce an inconsistent set of constraints or lead to wrong hypotheses [9]. Differentiable logical approaches [10] or the use of soft rules [11] are potential remedies to better deal with noisy data.
>
> **(iii)** It would probably be more precise to state that most embedding-based methods (e.g., TransE, DistMult, or ComplEx) have difficulties in capturing long-range dependencies since they only minimize the reconstruction error in the immediate first-order neighborhoods and are not explicitly considering higher-order neighbors during training [12]. Especially the expressiveness of long-tail entities might be low due to the small number of neighbors [13].
>
> **Q3**: Maybe I overlooked something, but I could not find the link to the source code in either the paper or the supplementary material.
>
> **A3**: The source code for the paper is now available at
> https://github.com/liu-yushan/eswc2021.
>
> References:
>
> [1] Hitzler, P. et al. Foundations of Sematic Web Technologies. 2009.
>
> [2] Galárraga, L. et al. Fast rule mining in ontological knowledge bases with AMIE+. The VLDB Journal 2015.
>
> [3] Wang, Z. and Li, J. RDF2Rules: learning rules from RDF knowledge bases by mining frequent predicate cycles. CoRR 2015.
>
> [4] Yang, F. et al. Differentiable learning of logical rules for knowledge base reasoning. NeurIPS 2017.
>
> [5] Das, R. et al. Go for a walk and arrive at the answer: reasoning over paths in knowledge bases using reinforcement learning. ICLR 2018.
>
> [6] Rocktäschel, T. and Riedel, S. End-to-end-differentiable proving. NeurIPS 2017.
>
> [7] Meilicke, C. et al. Anytime bottom-up rule learning for knowledge graph completion. IJCAI 2019.
>
> [9] Mitchell, T. Machine Learning. Chapter 10. 1997.
>
> [10] Evans, R. and Grefenstette, E. Learning explanatory rules from noisy data. IJCAI 2018.
>
> [11] An, B. et al. Retrofitting soft rules for knowledge representation learning. JIST 2019.
>
> [12] Hamilton, W. Graph Representation Learning. 2020.
>
> [13] Guo, L. et al. Learning to exploit long-term relational dependencies in knowledge graphs. ICML 2019.

---

### Official Review · AnonReviewer5 · 2021-01-15
**Good work on interesting problem**

**Rating:** 2
**Confidence:** 4
**Impact:** 3
**Design And Technical Quality:** 3

**Review:**

This paper presents a novel neuro-symbolic approach developed on the top of an existing reinforcement learning method (i.e., MINERVA) for solving the drug repurposing problem. The authors inject some background knowledge in the form of logical rules in the reinforcement learning space and change the reward function accordingly to guide the agent in taking the actions (selecting the next relation and node in its walk) using such background rules (meta paths information). The paper tackles an important and interesting problem in the biomedical domain. My main concern however is the methods that have been chosen in the experiments for the sake of comparison. While the authors have tried to cover multiple methods from symbolic, sub-symbolic and neuro-symbolic domains, they have not compared their result with Neural Theorem Provers(NTP). As the authors explained NTPs are not scalable but I suggest the authors compare their model with new more scalable variation of NTPs like Greedy NTPs (GNTPs) [1] or Conditional Theorem Provers (CTPs) [2]. I like to see either more experiments or acceptable justification for not comparing their model with the above mentioned models.

Overall, a study on the drug repurposing using neuro-symbolic link prediction approach is an interesting idea. The paper is also very well-written. I'd like to see the revised manuscript being accepted by the conference assuming the authors will resolve the raised issue in the rebuttal phase.

[1] Minervini, Pasquale, et al. "Differentiable Reasoning on Large Knowledge Bases and Natural Language." (2020): 125-142.
[2] Minervini, Pasquale, et al. "Learning reasoning strategies in end-to-end differentiable proving." International Conference on Machine Learning. PMLR, 2020.

Post-rebuttal: The rebuttal addresses the issue. Upgraded to accept.

**Anonymity:**

Yes, I would like my review to remain anonymous.

**Reuse And Availability:**

3: Medium

**Strong Points:**

* The paper is very well-written and crystal clear.
* Although the idea of background knowledge injection in the reinforcement learning agents is not novel in general; such studies show the potential of such models in practice.


**Subreviewer:**

I delegated this review to a subreviewer.

**Weak Points:**

More experiments are required to prove the better performance of the proposed method compared to its neuro-symbolic counterparts (see above).

---

> ### Author Rebuttal · Authors · 2021-01-30
>
> Thank you very much for your feedback and the pointers to the more scalable variations of NTP!
>
> After comparing the methods GNTP [1] and CTP [2], we decided to conduct more experiments with CTP. CTP is a more recent extension of NTP and learns an optimal rule selection strategy via gradient-based optimization, while GNTP considers the top-$k$ facts and rules that are most likely to prove the goal during reasoning. The authors show that CTP is able to produce more accurate link prediction results than baseline methods (including GNTP) while maintaining low complexity of the reasoning process [2].
>
> We use the implementation by the authors (https://github.com/uclnlp/ctp) and obtain the following preliminary results on Hetionet:
>
> Hits@1: 0.166
>
> Hits@3: 0.305
>
> Hits@10: 0.510
>
> MRR: 0.283
>
> For comparison, the results reported in our submission for MINERVA+ (MINERVA+ (pruned)) are:
>
> Hits@1: 0.314 (0.337)
>
> Hits@3: 0.428 (0.470)
>
> Hits@10: 0.609 (0.641)
>
> MRR: 0.402 (0.430)
>
> Due to the short duration of the rebuttal phase, we were only able to run few models/hyperparameter settings. We will do a more extensive hyperparameter sweep and include the results for CTP in the final version of the paper.
>
> [1] Minervini, P. et al. Differentiable reasoning on large knowledge bases and natural language. AAAI 2020.
>
> [2] Minervini, P. et al. Learning reasoning strategies in end-to-end differentiable proving. ICML 2020.

---

> > ### Comment · AnonReviewer5 · 2021-02-01
> > **issue resolved**
> >
> > Thank you this resolves the issue.

---

### Decision · Program_Chairs · 2021-02-23

**Decision:**

Accept

**Comment:**

This paper presents a novel neuro-symbolic approach developed on top of an existing reinforcement learning method (i.e., MINERVA) for solving the drug repurposing problem. The authors inject background knowledge in the form of logical rules in the reinforcement learning space and change the reward function accordingly to guide the agent in taking the actions. The work is novel, interesting, clearly written and fits the ESWC venue. As all of the reviewers are positive and are satisfied with the response of the authors, we suggest acceptance.

To address the comments of reviewers in the final version, the authors are asked to expand experiments, extend the discussion of the related approaches and add a link to the source code.